# How to Classify Pituitary Neuroendocrine Tumors (PitNET)s in 2020

**DOI:** 10.3390/cancers12020514

**Published:** 2020-02-22

**Authors:** Jacqueline Trouillas, Marie-Lise Jaffrain-Rea, Alexandre Vasiljevic, Gérald Raverot, Federico Roncaroli, Chiara Villa

**Affiliations:** 1University of Lyon 1, University of Lyon, F-69000 Lyon, France; alexandre.vasiljevic@chu-lyon.fr (A.V.); gerald.raverot@chu-lyon.fr (G.R.);; 2Faculty of Medicine Lyon-Est, F-69372 Lyon, France; 3Biotechnological and Applied Clinical Sciences Department, University of L’Aquila, 67100 L’Aquila, Italy; marielise.jaffrain@univaq.it; 4Neuroendocrinology, Neuromed Institute, IRCCS, 86077 Pozzilli, Italy; 5Cancer Research Center of Lyon (CRCL), INSERM U1052, CNRS UMR5286, University of Lyon, 69008 Lyon, France; 6Pathology department, Groupement Hospitalier Est, Hospices Civils de Lyon, 69677 Bron, France; 7Endocrinology Department, Groupement Hospitalier Est, Hospices Civils de Lyon, 69677 Bron, France; federico.roncaroli@manchester.ac.uk; 8Division of Neuroscience and Experimental Psychology, School of Biological Sciences, Faculty of Biology, Medicine and Health, University of Manchester, Manchester, Manchester M13 3PL, UK; federico.roncaroli@manchester.ac.uk; 9Pathological Department, Foch Hospital, 40 rue Worth Suresnes, 92151 Suresnes, France; cm.villa@hopital-foch.org; 10INSERM U1016, CNRS UMR 8104, Paris Descartes University, Cochin Institute, 24 rue du Faubourg Saint Jacques, 75014 Paris, France; 11Endocrinology Department CHU de Liège, University of Liège, Sart Tilman B35, 4000 Liège, Belgium

**Keywords:** pituitary adenoma, pituitary tumor, classification of pituitary neuroendocrine tumors, classification of pituitary tumors

## Abstract

Adenohypophyseal tumors, which were recently renamed pituitary neuroendocrine tumors (PitNET), are mostly benign, but may present various behaviors: invasive, “aggressive” and malignant with metastases. They are classified into seven morphofunctional types and three lineages: lactotroph, somatotroph and thyrotroph (PIT1 lineage), corticotroph (TPIT lineage) or gonadotroph (SF1 lineage), null cell or immunonegative tumor and plurihormonal tumors. The WHO 2017 classification suggested that subtypes, such as male lactotroph, silent corticotroph and Crooke cell, sparsely granulated somatotroph, and silent plurihormonal PIT1 positive tumors, should be considered as “high risk” tumors. However, the prognostic impact of these subtypes and of each morphologic type remains controversial. In contrast, the French five-tiered classification, taking into account the invasion, the immuno-histochemical (IHC) type, and the proliferative markers (Ki-67 index, mitotic count, p53 positivity), has a prognostic value validated by statistical analysis in 4 independent cohorts. A standardized report for the diagnosis of pituitary tumors, integrating all these parameters, has been proposed by the European Pituitary Pathology Group (EPPG). In 2020, the pituitary pathologist must be considered as a member of the multidisciplinary pituitary team. The pathological diagnosis may help the clinician to adapt the post-operative management, including appropriate follow-up and early recognition and treatment of potentially aggressive forms.

## 1. Introduction

For about a century, acromegaly and Cushing’s disease have been considered endocrine conditions of hypothalamic origin or due to a benign pituitary tumor named “adenoma”. The World Health Organization (WHO) classification published in 2017 [1] maintained the term “adenoma” as consecrated by use, although a sizeable proportion of tumors of adenohypophyseal cells invade the surrounding structures and cannot be cured with standard treatments. A minority of invasive tumors can be defined as “clinically aggressive” but only exceptional tumors (0.2%) that metastasize are defined as “carcinoma”. The results of a survey of the European Society of Endocrinology (ESE) showed that locally aggressive tumors and carcinomas share many similarities [2] and can be regarded as two sides of the same coin [3].

This review will present the classification of pituitary neuroendocrine tumors in 2020, focusing on the prognostic classifications.

## 2. From Adenoma to Pituitary Neuroendocrine Tumor (PitNET): Evolution Not Revolution

The last years have witnessed a debate on the nomenclature of tumors of adenohypophyseal cells. In 2017, the members of the International Pituitary Pathology Club proposed the use of the term “neuroendocrine tumor” rather than “adenoma” to define tumors of adenohypophyseal cells [4]. Some endocrinologists criticized the change from adenoma to PitNET [5], arguing that adenoma has a benign connotation while tumor may have a sinister connotation. It should be noted that about 40% of the pituitary tumors are invasive towards the cavernous sinus, and less commonly the bone, being a cause of morbidity in patients. “Invasive adenoma” is therefore an oxymoron. A panel of experts of the WHO/IARC also proposed to include pituitary tumors in “neuroendocrine neoplasms” (NEN), which divided into the neuroendocrine carcinomas (NEC) for aggressive, poorly differentiated tumors and the neuroendocrine tumors (NET) for the well differentiated, and generally low-grade neoplasms [6]. Although a large majority of PitNETs will behave as well differentiated, benign neoplasms, we believe that this terminology can only impact positively on clinical practice as it reflects more closely the variability of behavior of pituitary tumor (invasiveness linked to a higher risk of recurrence) and may open up new strategies for the early identification and management of the most aggressive forms. It is proven that the anxiety of patients, when exposed to new terminologies, can be dealt with by experienced physicians [7]. For such reasons, we have chosen to endorse the terminology of PitNET.

## 3. Clinical and Pathological Classifications of PitNETs

### 3.1. Clinical Classifications

PitNETs are divided into clinically functioning and non-functioning (NF). Functioning tumors cause conditions related to hormonal hypersecretion such as acromegaly due to elevated plasma growth hormone (GH) and/or insulin growth factor 1 (IGF1), amenorrhea-galactorrhea or hypogonadism due to hyperprolactinemia, or Cushing’s disease due to hypercortisolism. Non-functioning tumors do not cause signs and symptoms of hypersecretion, except hyperprolactinemia due to the hypothalamic disconnection [8,9]. Clinically, NF tumors belong mainly to the gonadotroph type and are immunoreactive for FSH and LH subunits [10]. They differ from the much rarer silent corticotroph [11], silent somatotroph [12], silent TSH [13] and “poorly differentiated monomorphous plurihormonal PIT1 lineage” tumors [14]. In the latter, the diagnosis is based on the expression of ACTH, GH, TSH or GH/PRL seen by immunohistochemistry (IHC). In some cases, Cushing’s disease or acromegaly may develop over time, suggesting a continuum between silent and functioning PitNETs [9,11].

### 3.2. Neuroimaging Classification

#### Tumor Size and Invasion

Based on Magnetic Resonance Imaging (MRI) data, PitNETs are classified into small (micro < 10mm), large (macro ≥ 10mm) and giant (≥40mm).

The percentage of invasive tumors of all types varied from 30% [15] to 65% [16], depending on cohorts, the study period, and diagnostic criteria. The percentage most frequently reported is 45% [17,18,19]. Pituitary tumors may invade one or both cavernous sinuses [20], and/or the bone and the respiratory mucosae breaching into the sphenoid sinus. The invasion of dura mater forming the diaphragm sellae does not associate to recurrence [17], and neither does suprasellar expansion [19,21]. Large tumors are often invasive. Tumor size and invasion are related, but only invasion is predictive of progression/recurrence [19,22,23].

### 3.3. Morphological and Functional Classification 

#### 3.3.1. Previous Classifications 

Based on their tinctorial properties of haematoxylin eosin and their correlations with clinical phenotype, pituitary tumors were classified into acidophilic, basophilic, or chromophobic tumors and correlated respectively with acromegaly, Cushing’s disease and absence of signs of hypersecretion. Later, the development of electron microscopy (EM) and IHC led to a classification based on the appearance of their organelles (such as granulations, mitochondria) and hormone expression respectively [24,25,26]. Taking into account the clinical phenotype, and ultrastructural and IHC features, the following morphofunctional subtypes were described: lactotroph or prolactin (PRL) adenoma or prolactinoma, somatotroph (GH) adenoma with both the sparsely (SG) and densely granulated (DG) ultrastructural subtypes, gonadotroph (FSH/LH) adenoma, with hypersecretion (review by [27] or with normal plasma levels [28,29], null cell adenoma [30], also termed immunonegative adenoma [26], and thyrotroph (TSH) adenoma [31,32].

#### 3.3.2. WHO 2004 Classification

The WHO classification published in 2004 [33] classified tumors into seven main types: GH-, PRL-, FSH/LH-, ACTH-, and TSH-producing lesions based on their immunoprofile (IHC for pituitary hormones), subdivided into 13 ultrastructural subtypes.

#### 3.3.3. Morphological and Functional Classification in 2020

The 2017 WHO classification [1] recognized the relevance of lineage-restricted pituitary transcription factors (TFs) to classify tumors of adenohypophyseal cells and divided them into the three following lineages: PIT1 (pituitary specific transcription factor 1), TPIT (pituitary cell restricted factor), SF1 (splicing transcription factor 1). Lactotroph, somatotroph and thyrotroph belong to the PIT1 lineage, corticotroph to the TPIT lineage and gonadotroph to the SF1 lineage. Null cell and plurihormonal tumors are separate groups. Subtyping based on EM (electron microscopy) was abandoned because EM is not in routine use.

#### 3.3.4. Lineage-Restricted Transcription Factors

The differentiation of normal pituitary cells during embryonic development depends on TFs. In brief, PIT1 drives the differentiation of somatotroph, lactotroph and thyrotroph cells and it is associated with ER (Estrogen receptor) and GATA2 (GATA transcription factor 2) for lactotroph and thyrotroph differentiation respectively. Corticotroph differentiation depends on TPIT and NeuroD1 (neurogenic differentiation), and the gonadotroph lineage depends on SF1 and GATA2 [34,35,36,37]. A recent study suggested the use of GATA3 as complementary to SF1 in the diagnosis of gonadotroph tumor [38].

The detection of TFs using IHC has not yet been fully validated. For example, the detection of TPIT in corticotroph tumors was not reproducible until the recent development of a new antibody [39]. Anti-SF1 antibodies can also give non-specific results and it is our experience that SF1 expression can be weak and focal in gonadotroph tumors. Detection of TFs is essential in tumors with only scattered immunopositive cells (≤5%), especially in silent corticotroph, somatotroph and plurihormonal tumors and in immunonegative tumors. Therefore, diagnosis based primarily or exclusively on TFs may be misleading [40], and since hormone expression in tumor cells remains crucial to clinical management, we believe that they should still be investigated first [41].

#### 3.3.5. Morphological Types and Subtypes

The great majority of lactotroph tumors is sparsely granulated. They show chromophobic cells that express PRL with a typical Golgian and less commonly diffused pattern. These tumors with PIT1 and ER. ER expression can also be quantified using a score from 1 to 12 [42]. It must be mentioned that some anti-ER antibodies used routinely in breast pathology do not always work in lactotroph tumors.

Somatotroph tumors were first differentiated using EM into DG, and SG tumors. Now they can be easily recognized by IHC using antibodies directed against low molecular weight cytokeratins (LMWCK). LMWCKs highlight perinuclear expression in DG and the characteristic fibrous bodies in SG. Transitional forms with overlapping patterns of cytokeratin are classified as intermediate type but they are clinically similar to DG [43]. Both subtypes express GH, PIT1, SSTR2 and SSTR5. In SG tumors, GH and SSTR2 expression can be weaker than in DG tumors [44,45]. Of note, monoclonal antibodies against SSTRs are now available, allowing obtaining reliable evidence of SSTR expression at the membrane level. The expression of SSTRs may then be quantified on a scale of 0 to 12, established by the percentage of positive cells x the intensity of the staining; a cut-off >5 was proposed to predict the response to somatostatin analogues (SA) [46]. Mixed somatolactotroph subtype, in which GH and PRL are expressed by different cells, are classified with the somatotroph type.

Corticotroph tumors are differentiated into sparsely or densely granulated. They typically show widespread cytokeratin expression and nuclear TPIT expression. ACTH expression varies among cases. It is usually lower in SG subtype, but can also be weak due to rapid release of the hormone or be affected by tissue processing. ACTH is in fact a small peptide that can easily leak outside tumor cells during fixation and paraffin embedding.

Gonadotroph tumors are composed of chromophobic cells. Their immunoprofile includes FSH and LH βsubunits, αSU, SF1 and GATA3 expression. Neoplastic cells may only express FSH -subunit or less frequently only LH -subunit. The diagnosis is confirmed by SF1 and GATA3 IHC, particularly in cases showing less than 5% FSH and/or LH positive cells. Rare tumors show predominant expression of the common αSU [10]. Of note, an anti-αSU specific antibody is no longer commercially available.

The rare thyrotroph tumors are often composed of fascicles of spindle cells and often contain calcification and fibrous connective tissue. A definitive diagnosis can only be reached with IHC. In addition to βTSH, they express PIT1, αSU and GATA2. Thyrotroph tumors are probably underdiagnosed due to the lack of specific commercial antibodies against βTSH. 

The WHO 2017 classification is presented in Table 1, with the types in italics, which will be discussed in the next paragraph.

#### 3.3.6. Plurihormonal Tumors

Plurihormonal tumors belong mainly to the PIT1 lineage. According to the EPPG algorithm and as shown in Table 2, they are classified into functioning and non-functioning tumors.

The EPPG (European pituitary pathology group) proposed the term of somatotroph plurihormonal PIT1-positive tumor to define lesions with acromegaly/gigantism that show variable expression of PIT1, GH, TSH and/or PRL and thyrotroph plurihormonal PIT1-positive tumors to define those presenting with central hyperthyroidism [13,47]. The EPPG (European Pituitary Pathology Group) argues that the clinical management of these tumors is different. The criteria for rare (0.9%) silent poorly differentiated PIT1-positive tumors [14], previously known as silent type 3 adenomas [48], are still poorly defined. Silent tumors with other combinations of pituitary hormone expression such as GH-ACTH or PRL-ACTH or ACTH-LH are exceptional [49,50].

Tumor encasing normal adenohypophyseal cells should not be misdiagnosed as plurihormonal. Distinction can be difficult when normal cells are in sizeable number. A special staining for reticulin fibres or the immunoreaction for collagen IV can highlight the residual reticulin fibres.

#### 3.3.7. Immunonegative Type Rather Than “Null Cell Adenoma”

The 2004, the WHO Classification [33] defined null cell adenoma as a “hormone immunonegative adenoma, with scattered cells that are immunopositive for glycoprotein hormones”. The vast majority of “null cell adenoma” express SF1 and therefore belong to the gonadotroph lineage. The concept of “null cell adenoma” was revisited in the 2017 WHO classification and defined as a tumor without IHC evidence of cell-type differentiation and negative for TFs. Such tumors are rare, about 1%–2% [8], but need to be specifically studied. Indeed, it has been recently claimed that “null cell adenoma” are more aggressive than gonadotroph adenoma [51,52]. We emphasize that immunonegative tumors should be diagnosed after ruling out primary non-pituitary neuroendocrine or metastatic sellar tumors (c.f. differential diagnosis paragraph).

Very rare types and subtypes are not discussed in detail in this review because they represent variants rather than distinct entities. Mammosomatotroph tumors, characterized by co-localization of GH and PRL in the same cells, are a variant of somatotroph tumors. In the WHO 2017 classification, acidophilic stem cell adenoma [53] has been mentioned in both the chapters on somatotroph tumor [54] and lactotroph adenoma [55]. In our view, acidophilic stem cell tumors are more likely to represent lactotroph tumors with abnormal and giant mitochondria [56]. Oncocytoma [57,58] account for a phenotypic variant characterized by abnormal accumulation of mitochondria but it should not be regarded as a separate entity. Oncocytomas more often belong to the gonadotroph lineage but any adenohypophyseal tumor can potentially accumulate mitochondria, probably as a result of ageing.

The classification we proposed, adapted from the WHO 2017 classification [1], and the algorithm recently proposed by the [41] is presented in Table 3.

#### 3.3.8. Difficult Differential Diagnosis

Sellar and parasellar structures can be the site of a broad range of tumors, some of which can be challenging to diagnose due to their rarity and morphological features [59]. In addition, adenohypophyseal cell tumors can present as a nasopharyngeal mass or present ectopically, without a clear anatomical relationship with the pituitary gland [60]. Different markers can be useful to determine the nature of these tumors [61,62,63,64].

Of note, very high Ki-67 index and high mitotic count should alert pathologists to consider other diagnosis than pituitary tumor. Indeed, Ki-67 exceeding 10% are rarely observed in pituitary tumor as shown in a recent cohort [65]. In particular, the diagnosis of null cell adenoma should be considered with caution these cases.

#### 3.3.9. Prognostic Classifications

##### “Atypical Adenoma” and “High Risk Adenomas” of the WHO 2004 and 2017 Classifications

The 2004 WHO classification introduced the concept of atypical adenomas to define tumors with uncertain malignant potential [33]. The diagnosis was “atypical adenoma required an elevated mitotic index and a Ki-67 labelling index greater than 3%, as well as extensive nuclear staining for p53 immunoreactivity”. The invasion is not mentioned as one of the criteria but atypical adenomas are very often invasive at onset. These criteria were considered too vague, leading to a wide range incidences from 2.9% [66] to 18.7% [67]. Its prognostic value was also regarded as weak due to the nature of follow-up data comparing typical and atypical adenomas [68,69,70].

In the 2017 WHO classification [1], the atypical adenoma is no longer considered. Instead, it has been suggested that the histotype itself may bear a prognostic value and that some subtypes such as male lactotroph tumors, silent corticotroph and Crooke cell adenoma, SG somatotroph tumor, and silent plurihormonal Pit1 positive tumor should be considered as “high risk” tumors due to their potential aggressive behaviour. However, the prognostic impact of histotype in predicting treatment response is not widely accepted.

For instance, according to a recent meta-analysis [71], there is no firm evidence of a more aggressive behavior of silent corticotroph tumors [72,73] despite the fact that they often present as macro- and invasive tumors [74,75]. Similar to changes observed in normal corticotroph in patients with hypercortisolism, Crooke cell tumors are characterized as the perinuclear deposition of cytokeratin filaments, resulting in peripheral displacement of the cytoplasm and hyaline appearance of cells. Although a few cases of Crooke cell tumor have been documented, there is no conclusive evidence that Crooke cell tumors are intrinsically more aggressive [76].

SG somatotroph tumors occur more frequently in young patients than DG ones and are more frequently large (86% vs. 58% in DG) and invasive (65% vs. 38%) tumors [43]. Expression of SSTR2 is also lower [45], but it is not proven that they behave more aggressively in terms of uncontrolled growth [77,78]. The aggressive behavior of silent somatotroph [79] and plurihormonal silent PIT1 [14] remains questionable. It should be considered that somatotroph tumors account only for a minority of aggressive/malignant tumors [2].

However, it has been proven that lactotroph tumors in males differ from those in females, being larger, more often invasive, and resistant to dopamine agonists. They are more often aggressive in males than in females, with a high risk of recurrence and malignancy. Moreover, the expression of ERα in men, lower than in women, is closely correlated to aggressiveness (Review in [80]). Some silent corticotroph tumors (72–77) are also more aggressive than corticotroph tumors with Cushing’s disease. Lactotroph and corticotroph tumors both represent the most frequent aggressive and malignant pituitary tumors [2]. For SG somatotroph tumors, and silent plurihormonal Pit1 positive tumors, there is not sufficient scientific evidence yet to consider that these subtypes are more aggressive than their countertype. However, the morphofunctional type must be taken into account in the pituitary classification. Indeed, if the five main types (somatotroph, lactotroph, thyrotroph, corticotroph and gonadotroph) are considered, the tumor type is statiscally correlated to the prognosis, but the risk of tumor progression of each type varies from one cohort to another [19,22,77]. So, the histotype is not per se a strong prognostic factor.

It has only been mentioned that Ki-67 index ≥ 3%, p53 expression, and the following French five-tiered prognostic classification “may be of prognostic significance”.

##### The French Five-Tiered Prognostic ClassificationCharacteristics of This Classification

In 2013, a French five-tiered prognostic clinicopathological classification was proposed [19] (review in [81,82,83,84]). This classification takes into consideration the tumor diameter, tumor type and grading (Figure 1). The grading is based on invasion and proliferation. The invasion was assessed at MRI. Only tumors Grade III and IV of Knosp’s classification were taken into account [20]. The suprasellar expansion [19,21], the invasion of dura mater forming the diaphragm sellae, is not associated with recurrence and neither is [17]. The invasion of bone and the respiratory mucosae breaching into the sphenoid sinus confirmed by histology were taken into account, but not the presence of normal pituitary surrounding the tumor (20–17). The proliferation was evaluated on the 3 proliferative markers (mitotic count, Ki-67 labelling index and p53).

The tumors were classified into five grades: grade 1a (non-invasive and non-proliferative); grade 1b (non-invasive and proliferative); grade 2a (invasive and non-proliferative); grade 2b (invasive and proliferative); and grade 3: metastatic tumor. In a multicentric retrospective [19,22] and a monocentric cohort from the Lyon cohort, the Grade 1a tumors were the most common (47.3% and 51.2% respectively) while grade 2b tumors represented 7%–8%.

The proliferation is considered when at least 2 of the 3 markers are present and over the cut-offs: Ki-67 ≥ 3%, mitotic count *n* > 2/10 HPF (High Power Field), and p53 positive. As shown in Figure 1, the tumors were classified into five grades: grade 1a (non-invasive and non-proliferative tumor); grade 1b (non-invasive and proliferative); grade 2a (invasive and non-proliferative); grade 2b (invasive and proliferative); and grade 3: metastatic tumor. In a multicentric retrospective and a monocentric cohort from Lyon cohort [19,22], the Grade 1a tumors were the most common (47.3% and 51.2% respectively) while grade 2b tumors represented 7%–8%.

##### Comments on the Criteria of the Grading

The inclusion of invasion in a pathological classification of pituitary tumors remains controversial as some argue that it is not in the remit of pathologists to comment on neuroimaging and intra-operative features [85,86,87]. In addition, evidence of invasion can rarely be seen microscopically (9%) [19]. Although we accept this criticism, invasion remains the main predicting factor of recurrence [22,77,85]. Considering the progress of neuroimaging and endoscopic surgery, parasellar and bone invasion can be reliably evaluated in the majority of the cases. The neurosurgeon is the most skilled to detect invasion and could provide this information with the surgical sample.

This classification is also based on proliferation, which is evaluated on the Ki-67 index, the mitotic count and the p53 positivity. The Ki-67 index has been used in pituitary tumor analysis since 1996 [88] and is routinely assessed in diagnostic practice. The Ki-67 is a non-histone nuclear and nucleolar DNA binding protein encoded by *MKI-67* gene on chromosome 10q26.2. The protein is highly conserved in vertebrates and is expressed in the G1, S and G2 phases of the cell cycle but not in the quiescent cells (reviewed in [89]). Ki-67 expression is assessed using IHC on paraffin embedded section and expressed as the percent positive cells against the overall number of tumor cells (labelling index). The Ki-67 labelling index correlates with patient outcomes in several tumor types including neuroendocrine tumors [90]. Various cut-offs, ranging from 1.3% [91] to 10% [92], which have sometimes been adapted to tumor subtype [93], have been proposed. The Ki-67 index is generally higher in functioning than non-functioning tumors [16]. A labelling index ≥ 3% has been suggested to have prognostic value [19,77,88], that is, the cut-off that is chosen in this classification. The prognostic role of p53 expression is debated. Method of quantification had not been established [94]. However, recent studies defined a positive p53 staining as >10 strongly labelled nuclei per 10 HPF) [19,66,91]. With the Ki-67 index alone being insufficient to predict tumor behavior, the mitotic count usually low in pituitary tumor and the method of quantification of p53 not well reproducible in all laboratories, a tumor was considered as proliferative if at least two of the three markers (Ki-67 ≥ 3%, mitotic count > 2/10HPF and p53 positive) were present.

##### Prognostic Value of the Five-Tiered Classification

Stratification in five grades was validated against a retrospective multicentric large cohort (410 patients) at eight-year follow-up [19] and in a prospective monocentric study on 374 patients [22] with at three-and-a-half-year follow-up. The clinical data (criteria of patient’s selection and exclusion, the time interval of MRI, the criteria of case and control in the case-control study, the cure and the progression/recurrence status, are detailed in the previous papers [19,22]. At three and a half or eight years after surgery, 30%–40% of the tumors had recurred after a complete removal, or progressed from a remnant, with the patients with immediate post-operative radiotherapy being excluded. Multivariate analyses of disease-free survival status and recurrence/progression revealed the significant prognostic value (*p* < 0.001) of age, tumor type, and grade across all tumors and for each tumor type. The risk of recurrence/progression at eight and three-and-a-half years was respectively 12-and 3.5-fold higher for grade 2b tumors compared to grade 1a. The prognostic value of this classification has been validated by two other studies, on 1470 patients in total [19,22,77,85].

## 4. Aggressive Tumor and Pituitary Carcinoma

According to the recent consensus of the European Society of Endocrinology, an aggressive tumor is clinically defined as a “large, radiologically-confirmed invasive lesion with an unusually rapid growth or clinically relevant growth or recurrence despite optimal standard therapies” [95]. However, the pre-operative growth rate is rarely documented unless serial MRI scans are performed. A non-invasive tumor, even with post-operative recurrence and a giant lactotroph tumor sensitive to dopamine agonist [96], is not considered aggressive. So, defined, the precise number of aggressive tumors remains unknown but it is estimated to be in the range of 10% in surgical series [94,97]. Grade 2b tumors defined according to the French classification bearing a high-risk of recurrence is estimated between 7.4% and 8.5% [22,77]. However, until now, the prevalence of grade 2b tumors, which are more likely to develop an “aggressive” behavior during the follow-up, is unknown.

Pituitary carcinomas are still defined by the occurrence of metastasis [98]. Their frequency is very low, accounting for about 0.1%–0.4% of pituitary tumors [77,99,100]. All pituitary carcinomas evolved from invasive macroadenoma and to our knowledge, there are no reports of metastasis at onset. Metastatic spread can be intra-axial, resulting from dissemination in the subarachnoid space or via lymphatic or blood vessels. 

Lactotroph and corticotroph carcinomas are the most common types, representing respectively 36% and 30% of published cases [2,99]. A phenotype change from a “silent” to a “secreting” tumor should be considered as a marker of aggressiveness and as a risk factor for malignancy [2,74]. The immunonegative tumors (TFs not tested) represent 7.5% of pituitary carcinomas compared to their lower incidence (2%) in surgical cohorts of all types of pituitary tumors [22]. There are no established histological criteria to diagnose a potential pituitary carcinoma on the primary tumors, but numerous mitoses, a Ki-67 ≥ 10% [92,101], often associated with p53 expression [102] should alert the pathologist about a possible malignant behavior. Neoangiogenesis has also been described in pituitary carcinoma [92,103].

These clinically “aggressive” tumors and carcinomas are similar statistically in terms of gender, patients’ age at onset, tumor type, Ki-67 index and p53 expression [3]. Mitotic count >2 is more frequent in carcinomas [2] but notably, 11 primary tumors of 14 that metastasized were grade 2b ([19] and Asioli S unpublished data).

## 5. Standardized Histological Report of Pituitary Tumors

In order to reach an accurate diagnosis of pituitary tumors, pathologists may either opt for a broad panel approach or for a more targeted diagnostic strategy guided by the patient’s clinical presentation. The EPPG has recently proposed a standardized report (Figure 2) for the diagnosis of pituitary tumors [41]. The diagnosis is mainly based on IHC applying the full panel of antibodies directed against pituitary hormones. IHC for TFs is proposed as a further step in cases of immunonegative tumors and/or in the presence of rare positive cells, as well as in unexpected hormone co-secretion (e.g. ACTH-PRL). ERα can be used for a lactotroph tumor, LMWCK, to help stratify somatotroph tumors and recognize corticotroph tumors and chromogranin A, synaptophysin and other markers, such as TTF-1, to help in the differential diagnosis of metastases and sellar non-neurendocrine tumors. The final diagnosis takes into account the morphological classification, the proliferation and the grading. For example: sparsely granulated somatotroph tumor, with proliferation (Ki-67: 5%, mitosis *n* = 3 and p53 positive). Grade 2b tumor according to the five-tiered prognostic classification.

The genetic, the molecular aspects and the pathogenesis of PitNET tumors are out of the spectrum of the present review. Molecular biology may help in identifying molecular markers of invasiveness and, perhaps, markers of malignancy. However, as molecular studies are currently expensive and not validated in pathology routine, they remain in the research domain. Readers interested in these aspects could refer to the review of one of the authors [104]. 

## 6. Conclusions

In 2020, PitNETs are classified into 7 main morphological and functional types, using IHC. Given the behavior of PitNETs, we strongly encourage the integration of clinicopathological parameters in a standardized report, as proposed by EPPG. The prognostic five-tiered classification, taking into account these parameters, may help the clinician to adapt the post-operative management of the patients, including appropriate follow-up and early recognition and treatment of potentially aggressive tumors. The pituitary pathologist must be considered as a member of the multidisciplinary pituitary team.

## Figures and Tables

**Figure 1 cancers-12-00514-f001:**
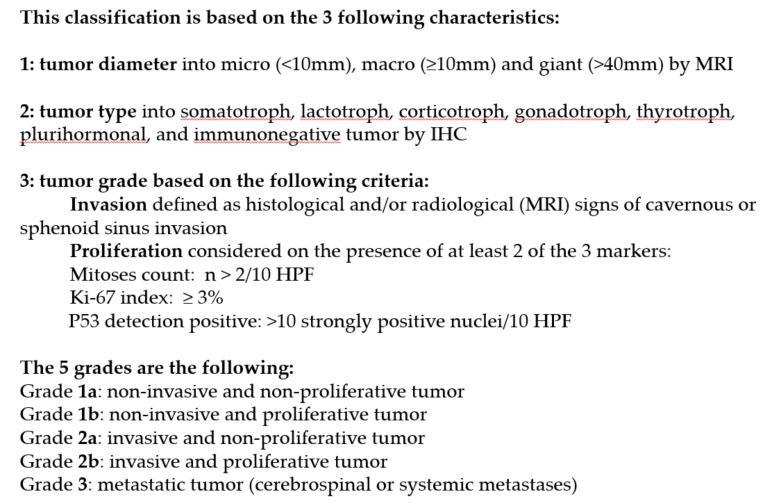
Five-Tiered prognostic classification of PitNETs [19].

**Figure 2 cancers-12-00514-f002:**
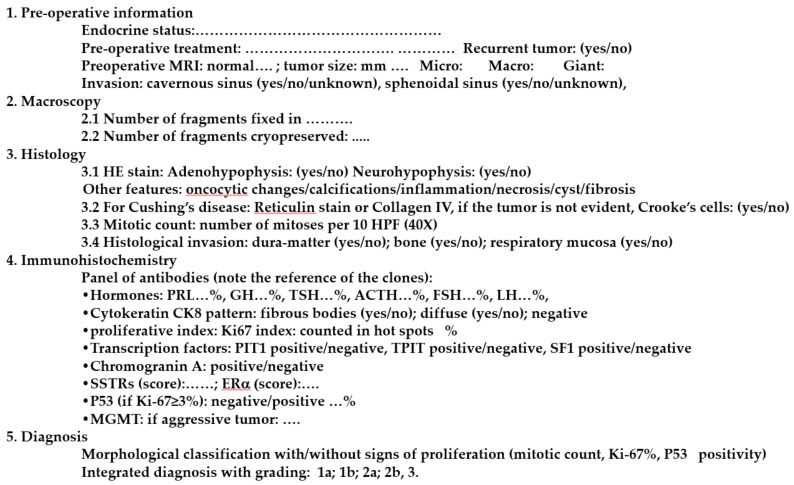
Standardized pathological report for PitNETs, adapted from EPPG [41].

**Table 1 cancers-12-00514-t001:** WHO 2017 Classification of Pituitary Adenomas, adapted from [1].

*Adenoma* Types		Immunophenotypes	Transcription Factors
Somatrotroph *Adenoma*	Densely granulated *adenoma*	GH ± PRL ± α-su LMW CK: diffuse	Pit-1
	Sparsely granulated *adenoma*	GH ± PRL LMW CK: fibrous bodies	Pit-1
	*Mammosomatotroph* adenoma	*GH + PRL* (in same cell) ± α-su	*Pit-1, ERɑ*
	Mixed Somatotroph-Lactotroph *adenoma*	GH + PRL (in different cells) ± α-su	Pit-1, ERɑ
Lactotroph *Adenoma*	Sparsely granulated *adenoma*	PRL	Pit-1, ERɑ
	*Densely granulated adenoma*	*PRL*	*Pit-1, ERɑ*
	*Acidophilic stem cell adenoma*	*PRL, GH* (focal and inconstant)	
Thyrotroph Adenoma		βTSH, α-su	Pit-1, GATA2
Corticotroph *Adenoma*	Densely granulated *adenoma*	ACTH, LMW CK: diffuse pattern	Tpit
	Sparsely granulated *adenoma*	ACTH, LMW CK: diffuse pattern	Tpit
	*Crooke’s cell adenoma*	ACTH, LMW CK: ring-like pattern	*Tpit*
Gonadotroph *adenoma*	Sparsely granulated	βFSH, βLH, α-su (various combinations)	SF-1, GATA2, ERɑ (variable)
*Null cell adenoma*		None	None
*Plurihormonal adenomas*	*Pit-1 positive plurihormonal adenoma (previously termed Silent subtype 3 adenoma)*	GH, PRL, βTSH ± α-su	*Pit-1*
	*Adenomas with unusual immunohistochemical combinations*	Various combinations	
Double adenomas	Distinct adenomas	Usually PRL and ACTH adenomas	Pit-1 and Tpit, respectively

GH: growth hormone; PRL: prolactin; TSH: thyrotrophine hormone; ACTH: adrenocorticotrophine hormone; FSH: folliculostrimulating hormone; luteotrophin stimulating hormone; LMW CK: low-molecular weight cytokeratin. In italic, subtype not used in the WHO 2020 classification Challenges posed by the 2017 WHO classification.

**Table 2 cancers-12-00514-t002:** Plurihormonal tumors of PIT1 lineage.

	Secretion	Immunophenotype	Type
**Functioning**	Acromegaly + Hyperthyroidism	GH±TSH ±PRL	Somatotroph plurihormonal
	Hyperthyroidism	TSH ±GH±PRL	Thyrotroph plurihormonal
	Variable	PRL-ACTH *	Double/triple tumors
**Non functioning**	No secretion	GH-TSH-PRL (Scarcely IR cells)	Silent plurihormonal/Poorly differentiated PIT1

* The most frequent.

**Table 3 cancers-12-00514-t003:** Classification and frequency of PitNET in 2020, adapted from the WHO 2017 classification [1] and the EPPG algorithm [41].

Tumor Types	Frequency %	Cytological Aspects	Immunophenotypes	Transcription Factors and Receptors
		Densely granulated	GH ± PRL ± α-subunit	PIT1, SSTR_2-3-5_
			LMW CK: diffuse	
**Somatotroph Tumor**	27	Sparsely granulated	GH ± PRL	PIT1 ± SSTR_2-3-5_
			LMW CK: fibrous bodies	
		Somatolactotroph	GH + PRL (in different cells) ± α-subunit	PIT1, ERɑ, ± SSTR
**Lactotroph Tumor**	21	Sparsely granulated	PRL	PIT1, ERɑ
**Thyrotroph Tumor**	2	Sparsely granulated	βTSH, α-subunit	PIT1, GATA_2_, SSTR_2-5_
**Corticotroph Tumor**	13	Densely granulated *	ACTH, βend;	TPIT, ± SSTR_5_
			LMW CK: diffuse	
		Sparsely granulated	ACTH, βend;	TPIT
			LMW CK: diffuse	
**Gonadotroph Tumor**	35	Sparsely granulated	βFSH, βLH, α-subunit	SF1, GATA_2_, ERɑ, ±SSTR
			(various combinations)	
**Immunonegative Tumor ****	2	Sparsely granulated	None	None
**Plurihormonal**		Various aspects	GH, PRL, βTSH ±	PIT1
**Tumor *****			α-SU	
**Double/Triple Tumors**	Exceptional	Distinct tumors	PRL and ACTH	PIT1/TPIT

GH: growth hormone; PRL: prolactin; TSH: thyrotrophine hormone; ACTH: adrenocorticotrophine hormone; FSH: folliculostrimulating hormone; luteotrophin stimulating hormone; LMW CK, low-molecular weight cytokeratin; * Most common subtype; ** Also termed null cell adenoma; *** see the different subtypes in Table 2.

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
