# Peer review of "How to Classify Pituitary Neuroendocrine Tumors (PitNET)s in 2020"

_cancers, 2020, doi:10.3390/cancers12020514_

Round 1

Reviewer 1 Report

Thank you for the opportunity to read and review this manuscript.

This is an extensive review of on how to classify neuroendocrine tumors of the pituitary by means immunohistochemistry and other traditional staining methods, supported by the clinical/biochemical presentation by 2020.

The manuscript also presents a five-tiered prognostic classification based on clinical and pathological criteria’s.

The manuscript, especially the first part is well written.

The background and the clinical and pathological classification of PitNETs is clear to me, however should of course also be evaluated by a pathologist.

The French five-tiered prognostic classification

This chapter describes a French protocol for prognostic evaluation, however I do still believe we are lacking the tools for differentiate the follow-up of patients with a remnant PitNEt after surgery.

 In my view the description of Ki-67 and p53 could be placed under the pathological classification chapter.

How to grade invasion the pitNETs should be more thoroughly described, we can leave this to the subjective evaluation of the different neurosurgeons.

Line 349-351 seems to belong to figure 2, and not the text.

The conclusion:

The first part of the conclusion is great.

I do agree that the five-tiered classification MAY help the clinicians in the post-operative care of the patients, however the prognostic value of this model still has to be evaluated in larger studies. Do not forget to mention that lactotrophic tumors in males and some corticotroph tumors represents more aggressive subtypes.-

Some small remarks:

Line 349-350, 351 is that a part of figure 2. It doesn’t fit into the text.

Figure 1 : Of the five grades only Grad 1a : non-invasive tumor is in the figure, please add the rest

In general abbreviations should be explained when used the first time.

Example:

Line 123 ERα GATA2, Line 124 Neuro D1.

PIT1, SF1, TPit.

Line 174 EPPG (explained in line 208)

abcam- Abcam Biochemicals

The author name should be removed line 80

Author Response

Thank you very much for your kind comments and your interest in our work

This chapter describes a French protocol for prognostic evaluation, however I do still believe

I am very happy that the reviewer asked me a lot of questions concerning our classification. However, to avoid to speak too much about our personal work, and because this review is focusing mainly on pathology, with the other authors, we propose to not give more details concerning the clinical protocol and to ask the clinical readers to read the 2 original papers (ref 19, 22) were the criteria of inclusion and the follow-up were explain in detail.

In my view the description of Ki-67 and p53 could be placed under the pathological classification chapter.

It could be a possibility, but we decide to maintain this paragraph in the prognostic classification with the new section “comments of the criteria of the classification” line 314 to 332

How to grade invasion the pitNETs should be more thoroughly described, we can leave this to the subjective evaluation of the different neurosurgeons.

The method of coting the invasion and the discussion concerning this important criteria are described lines: 283 and 308 to 313

Line 349-351 seems to belong to figure 2, and not the text.

Thank you to notice this editing mistake

I do agree that the five-tiered classification MAY help the clinicians in the post-operative care of the patients.

I thank to the reviewer for his/her remark

The prognostic value of this model still has to be evaluated in larger studies

The prognostic value of this classification has been evaluated by statistic analyses on one retrospective and one prospective studies (see details in original papers (19-22). Its prognostic value has been confirmed by 2 other independant cohorts. So the prognostic value of this classification has been yet evaluate on 1470 patients: 1 French cohort and 1 Lyon cohort: 410 + 374 patients + 574 patients (Asioli cohort : ref 77 ) +120 patients (lelotte cohort ref 85). I know that an other european study on 1030 patient is under review and confirm its prognostic value. Line 344-345. I will very happy, if the reviewer 3 tests this classification on his/her patients.

Do not forget to mention that lactotrophic tumors in males and some corticotroph tumors represents more aggressive subtypes.

It was mentioned, but briefly, because it was my favorite subjects. I coted only a recent review (80). I detailed more lines 261-265

Some small remarks:

Line 349-350, 351 is that a part of figure 2. It doesn’t fit into the text.

Thank you. I remove it line 436

Figure 1: Of the five grades only Grad 1a : non-invasive tumor is in the figure, please add the rest

Thank you. I did it line 303

In general abbreviations should be explained when used the first time.

I did it

Lines 116-117: PIT1, SF1, TPit.

Line 124-126: ERα, GATA2, Neuro D1.

Line 183 EPPG

Line 133 abcam- Abcam Biochemicals

The author name should be removed line 80

Thank you. It has been done on line 50 and 80

Reviewer 2 Report

The imaging quality of figure 2 should be improved by retyping the reports.

Author Response

Thank you very much to the reviewer. It is the first time that one of my manuscript take 4 stars for all items with only one important comment.

I improve the quality of figure 2 in retyping it

I did many changes in response to the other revieweres. I hope that you will agree

Reviewer 3 Report

The manuscript summarizes a number of previous publications of this group which obviously want to have their grading system implemented in daily practice. Therefore there are no new data but a good overview on the current histopathological diagnosis of pituitary adenomas and the new WHO classification, which can serve as a discussion paper for physicians acting in this field, who may find it useful in their daily practice or not. Some statements are really not useful such as that “in 2020, the pituitary pathologist must be considered as a member of the multidisciplinary pituitary team”. The pituitary pathologist has always been a member of the multidisciplinary team; this is really not a conclusion to be drawn from the data.

Figure 1: seems to be incomplete (the five grades are the following, but only grade1a is presented)

Figure 2: lines 349- 352 seem to be double (already in the figure)

The authors should describe how they perform the follow up according to their grading system (time interval of MRI, radiation/radiosurgery early after surgery necessary after surgery or not, watch and wait for how long in recurrence etc.). This would be helpful for the physicians treating patients with pituitary adenomas.

Author Response

I thank very warmly the reviewer for taking time to review this manuscript and his/her kind comments

This manuscript summarizes a number of previous publications of this group which obviously want to have their grading system implemented in daily practice.

Of course, we will very happy if this French classification will be used especially in US. Indeed as an expert of the WHO 2017 classification, I did not succeed to have a description of this classification and to change the term of adenoma to tumor!!!!

Some statements are really not useful such as that “in 2020, the pituitary pathologist must be considered as a member of the multidisciplinary pituitary team”. The pituitary pathologist has always been a member of the multidisciplinary team; this is really not a conclusion to be drawn from the data.

From this reviewer’s comment, I guess that in his/her group, the management of the patients with a pituitary tumor is multidisciplinary. Unfortunately it is not the case all over the world, especially in Europe. For example, 8 years ago, I created EUROPIT, a European multidisciplinary course on pituitaty tumors, with clinicians, neurosurgeons and endocrinologists. In their evalution (96% of satisfaction), and comments, the students wrote that « this course will change their clinical management ». Many of them never come in the operative theater and know nothing in pathology. So with the respect to the reviewer, I did not take off this sentence

Figure 1: seems to be incomplete (the five grades are the following, but only grade1a is presented).

I ask the editing team to insert the correct figure 1.

Figure 2: lines 349- 352 seem to be double (already in the figure).

I remove it

The authors should describe how they perform the follow up according to their grading system (time interval of MRI, radiation/radiosurgery early after surgery necessary after surgery or not, watch and wait for how long in recurrence etc.). This would be helpful for the physicians treating patients with pituitary adenomas.

I am very happy that the reviewer asked me a lot of questions concerning our classification. I gave more details in 2 special paragraphs. However, to avoid to speak too much about our personal work, and because this review is focusing mainly on pathology, with the other authors, we propose to not give more details concerning the clinical protocol and to ask the clinical readers to read the 2 original papers (19-22) were the criteria of inclusion and the follow-up and the criteria of cure and progression were explain in detail.

Round 2

Reviewer 1 Report

I am satisfied with the Authors response and the changes made in the manuscripte and have no further comments.